# Micronutrient Adequacy in the Diet of Reproductive-Aged Adolescent Girls and Adult Women in Rural Bangladesh

**DOI:** 10.3390/nu13020337

**Published:** 2021-01-23

**Authors:** Rumana Akter, Hiroaki Sugino, Nasima Akhter, Christopher L. Brown, Shakuntala H. Thilsted, Nobuyuki Yagi

**Affiliations:** 1Graduate School of Agricultural and Life Sciences, The University of Tokyo, Room 533, Agriculture Building 7-B, 1-1-1 Yayoi, Bunkyo-ku, Tokyo 113-8657, Japan; a-sugino@mail.ecc.u-tokyo.ac.jp (H.S.); yagi@fs.a.u-tokyo.ac.jp (N.Y.); 2Department of Anthropology, Durham University, Durham DH1 3LE, UK; nasima.akhter@durham.ac.uk; 3FAO World Fisheries University Pilot Programme, Sinseon-ro, Nam-gu, Busan 48547, Korea; brownchristopher38@gmail.com; 4WorldFish, Jalan Batu Maung, Batu Maung, Bayan Lepas 11960, Penang, Malaysia; s.thilsted@cgiar.org

**Keywords:** diet quality, micronutrient, reproductive-aged women, nutrient adequacy

## Abstract

Micronutrient deficiencies remain a serious nutritional concern in Bangladesh, especially among rural women of reproductive age (WRA). This study assesses the diet quality of reproductive-aged adolescent girls and adult women (referred to together as WRA in this study), including socio-demographic factors associated with their diet quality. The diet quality of adolescent girls was compared with that of adult women to assess which group was most at risk. The diet quality was measured by calculating the nutrient adequacy ratio (NAR), using the preceding 24 h dietary recall method. The mean adequacy ratio (MAR) was calculated as an overall measure of diet quality using the NAR. Nearly three quarters of WRA (adolescents: 73.1–88.5%; adult women: 72.9–86.4%) had an inadequate intake of calcium, vitamin A, folic acid, and vitamin B12. The prevalence of inadequate dietary intakes of calcium, zinc, and energy was significantly higher in adolescent girls (*p* < 0.001) than in adult women. Overall diet quality was significantly better in adult women (0.51 ± 0.21, *p* < 0.001) than in adolescent girls (0.49 ± 0.22). Age, marital status, educational level, and monthly household income were important factors associated with the diet quality of WRA. Micronutrient inadequacy is widely prevalent in the diets of WRA in Bangladesh, and adolescent girls with poor socio-economic status and lower educational levels are at higher risk.

## 1. Introduction

Micronutrient malnutrition is a serious nutritional concern globally, affecting more than two billion individuals, or about a quarter of the world’s population [1,2]. This problem is more severe in developing countries than in developed countries [3]. Adequate maternal micronutrient status is critical during pregnancy and lactation [4,5]. Maternal micronutrient deficiencies are associated with various adverse health outcomes of children, such as impaired cognition, growth retardation, birth defects, and increased risks of morbidity and mortality [6,7]. Multiple micronutrient deficiencies in Bangladesh remain high, especially among young children, adolescent girls, and reproductive-aged adult women, accounting for a $7.9 billion loss in the annual national GDP (GDP, Gross Domestic Production) [8]. Staple-based monotonous diets that lack diversity contribute substantially to the high rate of micronutrient deficiencies in Bangladesh [9,10,11]. Furthermore, an inadequate intake of animal-source foods (ASFs) has been recognized as one of the most important reasons for the poor diet quality and micronutrient inadequacy among people in rural Bangladesh [12]. Fish is the most commonly and frequently consumed ASF compared to other animal food sources such as meat, dairy, and eggs, but the quantity of intake is small [13,14,15]. Other animal food sources are consumed less frequently [4]. Common micronutrient deficiencies reported in Bangladesh are vitamin A, iron, calcium, folic acid, zinc, vitamin B12, and iodine deficiencies [8].

Considering the higher micronutrient deficiency among adolescent girls and reproductive-aged adult women in Bangladesh, precise data on their food intake behavior are necessary for the development of food-based dietary guidelines in order to break the cycle of intergenerational transmission of malnutrition (from mother to children). This effort will contribute to a reduction in poverty and food insecurity, thus helping achieve the Sustainable Development Goals (SDGs) [16,17]. Although the reproductive age of women is considered to be between the ages of 15 and 49 years [17], in this study, reproductive age is considered to be between the ages of 13 and 49 years (adolescent girls: 13–18 years; and adult women: 19–49 years) as the onset of menarche is associated with early marriage [18] and child-bearing among adolescent girls in rural Bangladesh [19]. Adolescent girls have critical nutritional needs due to rapid physical, cognitive and psychological growth and development from childhood to adulthood. Hence, when adolescent girls become pregnant, it affects their own health and well-being as well as that of their children, thereby continuing the intergenerational cycle of malnutrition [20].

This study assesses the diet quality of reproductive-aged women (WRA), through measuring nutrient adequacy in the diet using the preceding 24 h dietary recall method. This study also compared the diet quality of adolescent girls with that of adult women to investigate which group is more nutritionally vulnerable. Socio-demographic factors associated with the diet quality of WRA were also studied. Particular nutrients of interest in this study, in addition to energy, are iron, calcium, zinc, vitamin A, thiamine, riboflavin, niacin, vitamin B6, folate, vitamin B12, and vitamin C. These micronutrients reflect key dimensions of diet quality [21] and are considered as the “nutrients of concern” globally [7,9,22].

## 2. Materials and Methods

This is a secondary analysis of a publicly available database, obtained from the Bangladesh Integrated Household Survey (BIHS) in 2015 [23] conducted by the International Food Policy Research Institute (IFPRI). Data were collected between January and June 2015 following a two-stage stratified sampling method, comprising 5447 households and 21,532 individuals and statistically representative of rural Bangladesh [24]. Details of the sampling and survey methodology have been published elsewhere [25]. Verbal consent was obtained from all the participants prior to their enrollment in the study.

### 2.1. Dietary Data Collection

Most of the enumerators had a Master’s degree in nutrition, social science, or home economics. The enumerators and their supervisors received extensive training in how to carry out the interviews, including a line-by-line explanation and interpretation of the questionnaires, flow and skip patterns, and definitions, as well as explanations of how to handle unusual cases and when to contact the supervisors for assistance. Field supervisors received additional training particularly related to the quality control process, such as cross-checking, editing, and coding the questions and security and confidentiality issues. The training of the survey team consisted of formal classroom-based training as well as closely monitored field practice.

Household and intra-household-level dietary intake data were collected from the person primarily responsible for preparing and serving food using the 24 h dietary recall method [26]. The following data were collected at the household level: the name and weight of the cooked foods/mixed dishes and the ingredients used to cook the foods/mixed dishes. For each individual in the household, the following information was collected: the name of foods/mixed dishes and the weight of each food/dish the individual had consumed. Therefore, the following calculation was used to obtain the equivalent of raw ingredients consumed by an individual in the household:$$\mathrm{Raw}\;\mathrm{weight}\;\mathrm{of}\;\mathrm{ingredients}\;\left(\mathrm{individual}\right)=\;\frac{\mathrm{raw}\;\mathrm{weight}\;\mathrm{of}\;\mathrm{ingredients}\;\left(\mathrm{household}\right)\times \;\mathrm{cooked}\;\mathrm{weight}\;\mathrm{of}\;\mathrm{consumed}\;\mathrm{foods}/\mathrm{mixed}\;\mathrm{dishes}\;\left(\mathrm{individual}\right)\;}{\mathrm{cooked}\;\mathrm{weight}\;\mathrm{of}\;\mathrm{foods}/\mathrm{mixed}\;\mathrm{dishes}\;\left(\mathrm{household}\right)}.$$

If any individual or household had an atypical meal in the preceding 24 h period due to special occasions such as festivals, weddings, or fasting, this was excluded from the dietary analysis. Pregnant or lactating women were also excluded during the dietary analysis.

### 2.2. Nutrient Database

To calculate nutrient intakes for individual WRA, raw ingredients consumed by individual WRA were converted to nutrients using a food composition database. The main source of nutrient data for this analysis was the most recent food composition database of Bangladesh [27]. However, since the Bangladeshi food composition database does not have nutrient information for all of the food items and mixed dishes consumed by the study participants, a new food composition database was compiled for this study using the Indian food composition table [28]; the Nepal food composition table [29]; and the United States Department of Agriculture (USDA) National Nutrient Database for Standard Reference, Legacy Release, April 2018 [30]. Thereafter, in order to assess individual dietary nutrient intakes, all the food items and mixed dishes consumed by the WRA were converted to nutrients using the compiled food composition database.

### 2.3. Calculating the Daily Dietary Nutrient Intakes of WRA

The dietary micronutrient intake of the WRA was compared with the age- and sex-specific estimated average requirement (EAR) [31] in order to calculate nutrient adequacy in the diet of WRA. An exception was made in the case of calcium. Since the EAR of calcium is unavailable, we have divided the recommended dietary allowances of calcium by 50 percent in order to get the EAR of calcium (EAR of calcium = RDA of calcium × 0.5/0.975). The daily energy requirement of WRA was calculated following the methodology Food and Agriculture Organization of the United Nations (FAO) [32], using the equation for basal metabolic rate (estimated from an individual’s age, body weight, and gender) and daily main physical activity level (PAL).

### 2.4. Measuring the Diet Quality of WRA

The mean adequacy ratio (MAR) was calculated as an overall measure of diet quality [26], using the nutrient adequacy ratio (NAR), in accordance with previous studies [33,34,35]. The NAR for a given nutrient is the ratio of an individual’s intake to the age- and sex-specific EAR [26,36]. NAR values were truncated at 1 so that a nutrient with an NAR greater than 1 could compensate for a nutrient with a lower NAR. The MAR is calculated by averaging all the truncated NAR values together, as described in Equation (2). Thus, the MAR is reported on a scale from 0 to 1, with 0 indicating that the requirement for no nutrients was met, and 1 indicating that the requirements for all nutrients were met.
(1)$$\mathrm{NAR}=\frac{\mathrm{Daily}\;\mathrm{nutrient}\;\mathrm{intake}}{\mathrm{EAR}},$$
(2)$$\mathrm{MAR}=\frac{\sum \mathrm{NAR}\;\left(\mathrm{each}\;\mathrm{truncated}\;\mathrm{at}\;1\right)}{\mathrm{Number}\;\mathrm{of}\;\mathrm{nutrients}}.$$

### 2.5. Statistical Analysis

Data were analyzed using the statistical software Stata (Version 15.1). All of the intake data and most of the socio-demographic data were not normally distributed. Therefore, descriptive statistics for continuous variables are presented as the median and range (minimum, maximum), and percentages (%) are used for categorical variables. An unpaired Student’s t-test was used to test the mean differences in the MAR between RA adolescent girls and adult women. Pearson’s chi-square test (χ^2^) was performed to test the differences in intake prevalence between both groups of women for the NAR of all nutrients. A *p* value of less than 0.05 was considered to be statistically significant. Pregnant or lactating women were excluded during the dietary analysis.

To account for the occasions in which more than one woman was recruited from the same household, a generalized estimating equation (GEE) was used to examine the associations of the MAR with the relevant individual- (age, marital status, level of education, occupation) and household-level factors (household size, total monthly income of the household). Variables that showed significant bivariate associations were included in the model. Sampling weights provided by the IFPRI were applied to all the results presented here.

## 3. Results

### 3.1. Basic Characteristics of the Households and WRA

The median number of people in each household was four (Table 1), and the total monthly household income was BDT 6250 (USD, United States Dollar $ 74.40) (XE currency converter, available online at https://www.xe.com/currencyconverter/ (accessed on 13 December 2018)). The majority of the WRA were married (65.4%), followed by unmarried (28.2%) and divorced/widowed/separated (6.5%). About one quarter (26%) of the WRA (13–49 years) had no formal education. Nearly two fifths of the WRA (42.3%) had formal education up to a secondary level, followed by those educated up to a primary level (27.4%) and up to college-level (3.6%).

Very few WRA had education up to the graduate level (0.7%). The majority (68.7%) of the WRA were not involved in any earning-related activities; fewer than one quarter (21.1%) of the WRA were farming either on their own farm or as sharecroppers (tenant farmers) or raising fish/poultry/livestock as their main occupation.

### 3.2. Dietary Nutrient Intakes of WRA in the Preceding 24 h

Table 2 shows that the dietary intakes of calcium, vitamin A, folic acid, and vitamin B12 were significantly (*p* < 0.001) lower than the age-and sex-specific EAR of adolescent girls in the preceding 24 h (intake vs. EAR: 300.5 vs. 666.7 mg/day, 128. vs. 452.5 μg/day, 152.6 vs. 290 μg/day, and 0.8 vs. 1.8 μg/day, respectively). Similar results were found in adult women (intake vs. EAR: 323.6 vs. 512.8 mg/day, 126.7 vs. 500 μg/day, 168.7 vs. 320 μg/day, and 0.8 vs. 2.0 μg/day, respectively). In contrast, dietary energy intakes were significantly (*p* < 0.001) higher than the requirement in both groups of WRA (intake vs. requirement: adolescent girls: 2001.3 vs. 1842.1 kcal, and adult women: 2208.5 vs. 1852.7 kcal).

### 3.3. Nutrient Inadequacy in the Diet of Adolescent Girls and Adult Women

Nearly three quarters of adolescent girls had inadequate dietary intakes of calcium, vitamin A, folic acid, and vitamin B12 (73.1–88.5%) in the preceding 24 h (Table 3). A similar intake pattern was found in adult women (72.9–86.4%). About two thirds of the WRA had an inadequate dietary intake of riboflavin (adolescent girls, 59.6%; adult women, 66.7%). Although median energy intake was higher than the requirement in both groups of WRA, about two-fifths of adolescent girls (39.6%) and more than one-quarter of adult women (27%) were energy deficient. The prevalence of inadequate dietary intakes of calcium, zinc, and energy was significantly higher in adolescent girls (*p* < 0.001) than in adult women.

### 3.4. Socio-Demographic Factors Associated with Diet Quality of WRA

Table 4 shows that adult women had a significantly better-quality diet than that of adolescent girls (*B* = −0.041, *p*<0.001). The educational levels of the WRA also influenced their diet quality; those who had up to primary (*B*= 0.023, *p* < 0.001), secondary, or higher-level education (*B* = 0.020, *p* < 0.005) had a significantly better-quality diet than those who had no education. Married women had a significantly higher MAR (*B* = 0.032, *p* < 0.05) than those who were unmarried or divorced/separated/widowed. Household monthly income was significantly (*p* < 0.05) associated with better diet quality.

## 4. Discussion

The median dietary intake of calcium, vitamin A, riboflavin, folic acid, and vitamin B12 was significantly lower than the EAR in both adolescent girls and adult women in the preceding 24 h. Nearly four quintiles of WRA had an inadequate dietary intake of vitamin A in the preceding 24 h. These results are consistent with other studies at a sub-sample level in Bangladesh [7,9,37]. Vitamin A is essential for normal visual function, growth, and development. Women need additional vitamin A during pregnancy to sustain fetal growth and development. A lower intake of vitamin A by adolescent girls can put them at greater risk during pregnancy. Lactating women also need more vitamin A to supply it to their young child through breast milk as well as for their own health [36].

Nearly three quarters of the WRA had an inadequate dietary intake of calcium, and more than half of them had an inadequate intake of riboflavin in the preceding 24 h. The prevalence of inadequate calcium intake was significantly higher in adolescent girls than in adult women. The higher prevalence of inadequate dietary intakes of calcium and riboflavin could be explained by the study results revealing inadequate dairy and egg consumption by women in rural Bangladesh [9,37]. Although the median intake of iron exceeded the EAR in both groups of women, more than a quarter of the WRA did not consume an adequate amount of dietary iron in the preceding 24 h. Iron deficiency is associated with the premature birth, impaired cognition, and reduced learning capacity of young children, which cannot be reversed by providing iron at a later stage [38].

The intake of dietary folic acid and vitamin B12 was far below the EAR in both groups of WRA. The prevalence of inadequate dietary intakes of folic acid was particularly higher in both of these groups. More than four quintiles of the WRA had an inadequate dietary intake of folic acid, and nearly three quarters of them had an inadequate vitamin B12 intake in the preceding 24 h. These results are consistent with studies conducted in rural Bangladesh showing inadequate animal-source food intake as one of the contributing factors, particularly for lower dietary vitamin B12 intake [9,39], as animal foods are the only source of vitamin B12. Although fish is the most commonly consumed and preferred animal-source food in Bangladesh, the quantity of intake is minimal [9,39]. B vitamins play important roles in the normal functioning of our body; particularly, folic acid and vitamin B12 deficiencies are associated with adverse maternal and neonatal outcomes [40]. A particular concern for these women of childbearing age is the lack of adequate folic acid in their diets, which is associated with neural tube defects [41]. Median energy intakes were higher than the requirements in both groups of WRA, which is consistent with studies in which a heavy reliance on rice by rural Bangladeshi people has been reported [7,11].

Overall, diet quality was significantly better in adult women than in adolescent girls. Younger age, level of education, and marital status were important factors associated with their diet quality. Younger WRA with lower education levels might not have decision-making and negotiation power within their household and control over their food choices. These results are consistent with studies in Bangladesh and other countries, showing that women’s education and decision-making power are associated with dietary diversity and diet quality in their households [42,43]. Higher monthly household income is significantly associated with better diet quality, although the gradient is small. Higher household income may have contributed to better diet quality through the increased food purchasing capacity of those households, which is supported by studies concluding that people tend to diversify their diets with increased household income [44,45,46].

Poor dietary practices lead to micronutrient deficiencies, which have detrimental impacts during pregnancy and untoward birth outcomes [22,47]. Given multiple micronutrient deficiencies among WRA in Bangladesh, the government of Bangladesh distributes free iron-folic acid and vitamin A supplementation nationwide, targeting WRA. However, these target-specific supplementations are highly resource-intensive and expensive; they are viewed as unsustainable. Moreover, a singular focus on one particular deficiency can leave other deficiencies unmet. The major deficiencies identified in this study will not be alleviated by programs or projects narrowly focusing on one or a few nutrients. Sustainable solutions are needed using locally available and culturally acceptable food systems that people can readily access. Dietary intervention programs and/or projects targeting WRA need to promote target-specific, nutrient-rich foods that are deficient among them. Further studies are recommended in order to understand the basic, underlying, and immediate causes of inadequate nutrient-rich food intakes of WRA in rural Bangladesh.

Various studies have concluded that many of the small indigenous fish species that are typically consumed whole in Bangladesh are a rich source of multiple vitamins and minerals (e.g., calcium, iron, zinc, vitamin A, folic acid, and vitamin B12) that are highly deficient among Bangladeshi people [48]. These small indigenous fish species have the potential to contribute to reducing multiple vitamin and mineral deficiencies through providing a highly bioavailable animal-source food [49,50,51,52,53].

## 5. Conclusions

The dietary intake of calcium, vitamin A, folic acid, and vitamin B12 was far below the EAR in both groups of WRA. Nearly three quarters of the WRA had an inadequate intake of calcium, vitamin A, folic acid, and vitamin B12 in the preceding 24 h. More than a quarter of them had inadequate dietary iron intakes. The prevalence of inadequate dietary nutrient intake was significantly higher in adolescent girls than in adult women for calcium, iron, and energy. The overall diet quality was significantly better in adult women than in adolescent girls. Younger age, marital status, lower level of education, and household monthly income were important factors associated with variability in the diet quality of WRA. The results of this study have significant policy implications in developing food-based dietary guidelines for WRA in rural Bangladesh.

## Figures and Tables

**Table 1 nutrients-13-00337-t001:** Basic characteristics of the surveyed households and WRA.

	Median (Range) ^1^	n
Household size	4 (1, 14)	5447
Household monthly income (BDT) ^2^	6250 (30, 140,000)	5447
Age of first given birth	17 (12, 36)	3975
Marital status of WRA	%	
Unmarried	28.2	
Married	65.4	4996
Divorced/separated/widowed	6.5	
Education levels of WRA		
No formal education	26	
Primary level (grade 1–5)	27.4	
Secondary level (grade 6–10)	42.3	4996
College (grade 11–12)	3.6	
Graduate	0.7	
Occupation of WRA		
Daily wage labor	4.6	
Monthly salaried work	2.4	
Self-employed	1.5	
Trader ^3^	0.8	4996
Production ^4^	0.9	
Farming	21.1	
Non-earning activities ^5^	68.7	

WRA, women of reproductive age; BDT, Bangladeshi currency; ^1^ values are median (minimum, maximum); ^2^ Bangladeshi currency (1 USD$ = 80 BDT); ^3^ small store, roadside stall, large shop, fish trader; ^4^ food processing, handicrafts, small industry; ^5^ students, housewife, physically/mentally challenged.

**Table 2 nutrients-13-00337-t002:** Dietary nutrient intakes of adolescent girls and adult women in the preceding 24 h.

	Median (Range) ^1^	EAR	MAR (Mean ± SD)
Adolescent girls ^3^ (*n* = 1110)			
Energy (kcal)	2001.3 (523.3, 6610.7)	1842.1 ^2^**	
Calcium (mg)	300.5 (10.6, 5180.1)	666.7 **	
Iron (mg)	9.2 (1.02, 123.9)	6.8	
Zinc (mg)	9.3 (1.4, 43.1)	7.2	
Vitamin A (μg)	124.8 (0.7, 4247.3.1)	452.5 **	
Thiamine (mg)	1.0 (0.1, 5.8)	0.8	0.49 * ± 0.22
Riboflavin (mg)	0.6 (0.1, 5.1)	0.9 **	
Niacin (mg)	17.9 (2.5, 96.3)	10	
Vitamin B6 (mg)	1.4 (0.1, 6.7)	0.9	
Folic acid (μg)	152.6 (15.9, 1049.3)	290 **	
Vitamin C (mg)	66.7 (0.6, 1162.4)	47.5	
Vitamin B12 (μg)	0.8 (0, 63.9)	1.8 ***	
Protein	55.7 (13.4, 159.5)		
Fat	25.5 (1.0, 114.3)		
Carbohydrate	364.2 (76.0, 758.9)		
Adult women ^4^ (*n* = 3,886)			
Energy (Kcal)	2208.5 (1000.8, 6697.8)	1852.7 ^2^ **	
Calcium (mg)	323.6 (9.88, 5243.5)	512.8 **	
Iron (mg)	10.8 (0.3, 99.0)	8.1	
Zinc (mg)	10.3 (0.5, 41.2)	6.8	
Vitamin A (μg)	126.7 (2.1, 6708.8)	500 **	0.51 ± 0.21
Thiamine (mg)	1.1 (0.3, 13.2)	0.9	
Riboflavin (mg)	0.7 (0.1, 5.4)	0.9 **	
Niacin (mg)	19.9 (4.3, 72.3)	11	
Vitamin B6 (mg)	1.7 (0.2, 7.8)	1.1	
Folic acid (μg)	168.7 (38.1, 1216.2)	320 **	
Vitamin C (mg)	72.8 (1.4, 1655.9)	60	
Vitamin B12 (μg)	0.8 (0, 66.8)	2.0 **	
Protein	62.0 (49.3, 77.2)		
Fat	27.3 (17.4, 40.7)		
Carbohydrate	402.1 (330.3, 481,8)		

EAR, estimated average requirement; MAR, mean adequacy ratio; SD, standard division;^1^ values are median (minimum, maximum); ^2^ daily dietary energy requirement; ^3^ age between 13 and 18 years; ^4^ age between 19 and 49 years; * significantly different in MAR between adolescent girls and adult women, * *p* < 0.001; ** *p* < 0.001; *** *p* < 0.05. MAR (mean ± SD) of 11 micronutrients and energy was significantly lower in adolescent girls than in adult women (adolescent girls, 0.49 ± 0.22; adult women, 0.51 ± 0.21; *p* < 0.001).

**Table 3 nutrients-13-00337-t003:** Prevalence of inadequate dietary nutrient intakes in adolescent girls and adult women.

Nutrients	Adolescent Girls (%) (13–18 years)	Adult Women (%) (19–49 years)	*p* Value
Energy	39.6	27.0	<0.001
Calcium	77.8	72.9	<0.001
Iron	28.4	26.3	0.098
Zinc	24.0	11.5	<0.001
Vitamin A	78.7	77.4	0. 567
Thiamine	38.1	36.5	0340
Riboflavin	59.6	66.7	0.072
Niacin	13.2	11.8	0.206
Vitamin B6	19.0	18.5	0.688
Folic acid	88.5	86.4	0.077
Vitamin C	37.7	40.5	0.090
Vitamin B12	73.4	73.1	0.962

**Table 4 nutrients-13-00337-t004:** Socio-demographic factors associated with the diet quality of WRA using GEE.

Parameter	Coefficient	95% CI	*p* Value
Intercept	0.512	0.490, 0.534	
Adolescent girls	−0.041	−0.057, −0.024	<0.001
Adult women	Reference		
Educational level			
Secondary level or above	0.020	0.006, −0.034	<0.005
Up to Primary level	0.023	0.010, 0.036	<0.001
No education	Reference		
Marital status			
Unmarried	0.021	−0.005, 0.047	0.205
Married	0.032	0.011, 0.053	0.002
Divorced/separated/widowed	Reference		
Household monthly income	0.000	0.000	0.015

WRA, women of reproductive age; GEE, generalized estimating equation; CI, confidence interval.

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
