# Peer review of "Micronutrient Adequacy in the Diet of Reproductive-Aged Adolescent Girls and Adult Women in Rural Bangladesh"

_nutrients, 2021, doi:10.3390/nu13020337_

Round 1
Reviewer 1 Report
- Write Line 54
- Why did you include adolescents as women of reproductive age explain?
- Remove line 82 since this is a secondary analysis
- Section 2.1; Dietary data collection is unclear?
- List inclusion and exclusion criteria
Analyze
- Reconstruct Table 1 to include only the sample of adolescent girls and adult women used for the study. Children and Men should be taken out and focus on only WRA. The sample size should be the same on the table. Include marital status, parity or number of children, BMI in Table 1. Mean and ranges for each household characteristics should be included.
- Include the age range of adolescent girls and adult women as a footnote in Table 2.
- Compare the nutrient intakes of adolescent and adult girls side by side with p-values and indicate which nutrients are different between the 2 groups.
- It will be beneficial to know the prevalence of multiple micronutrient deficiencies among this population since the prevalence of most nutrients are high.
Discussion section
- Start this section by talking about the general MN deficiency found in this population. It is too abrupt to talk about calcium in the beginning paragraphs.
- Then focus on MN of public health importance
- Discussions can be tightened to reflect its implication for WRA and propose other studies and policies that can be adopted in Bangladesh.
- Include strengthen and weakness of the study
Author Response
Responses to reviewer-1
Thank you so much for the time in reviewing the manuscript.
Comment 1: Write Line 54
Response 1: We have revised line 54 (now lines 55-59)
Comment 2: Why did you include adolescents as women of reproductive age explain?
Response 2: Thank you for asking this question. As we have mentioned in lines 60-66 that onset of menarche is associated with early marriage and child-bearing among adolescent girls in rural Bangladesh. Hence, when adolescent girls become pregnant, it affects their own health and well-being as well as that of their children, thereby continuing the intergenerational cycle of malnutrition. Therefore, considering country context, this study included adolescents in the reproductive age and assessed their diet quality.
Comment 3: Remove line 82 since this is a secondary analysis
Response 3: Thank you for the comment. We have revised the line. Other reviewers have suggested to add more information regarding enumerators training and data collection for the readers. Therefore, we had to expand the section further (lines 84-91).
Comment 4: Section 2.1; Dietary data collection is unclear?
Response 4: Thank you for pointing this. We have revised this section with further details information in lines 92-101. Hope it is more clearer now.
Comment 5: List inclusion and exclusion criteria
Response 5: Thank you for the suggestion. Study exclusion criteria has been added in lines 103-105.
Comment 6: Reconstruct Table 1 to include only the sample of adolescent girls and adult women used for the study. Children and Men should be taken out and focus on only WRA. The sample size should be the same on the table. Include marital status, parity or number of children, BMI in Table 1. Mean and ranges for each household characteristics should be included.
Response 6: Thank you for the comments. We have removed information related to children and men from the Table 1 as advised. Marital status of WRA is also included to the table 1.We do not have information of number of children of WRA in the database (only have family size). Since the focus of our study is micronutrient intakes, we have not analyzed anthropometry data. Due to the skewed data, we have provided median and ranges of household characteristics, such as income and household size.
Comment 7: Include the age range of adolescent girls and adult women as a footnote in Table 2.
Response 7: Thank you for the suggestion. We have added the age range of adolescent girls and adult women as a footnote in Table 2 as suggested.
Comment 8: Compare the nutrient intakes of adolescent and adult girls side by side with p-values and indicate which nutrients are different between the 2 groups.
Response 8: Thank you for the comment. As you know that authors have compared prevalence of inadequate dietary nutrient intakes and diet quality between adolescent girls and adult women side-by-side with p values in table 3. Comparing their mean intakes between two groups may not be very useful as their requirements and cut-points are different.
Comment 9: It will be beneficial to know the prevalence of multiple micronutrient deficiencies among this population since the prevalence of most nutrients are high.
Response 9: Thank you for the comment. At this point we have only looked at the key nutrients that are considered as “nutrients of concern” globally. We also do not have food composition database for many nutrients.
Comment 10: Start this section by talking about the general MN deficiency found in this population. It is too abrupt to talk about calcium in the beginning paragraphs.
Response 10: Thank you for the suggestion. We have revised the discussion part thoroughly as suggested.
Comment 11 & 12: Then focus on MN of public health importance. Discussions can be tightened to reflect its implication for WRA and propose other studies and policies that can be adopted in Bangladesh.
Responses 11 & 12: We have revised the entire discussion part along with recommendation for further studies and interventions in lines 260-266 and policy implications in lines 281-282.
Comment 13: Include strengthen and weakness of the study
Response 13: Thank you for the suggestion. Strengths and weakness are added in lines 288-294.

Reviewer 2 Report
Please refer to attached PDF file.

Author Response
Responses for reviewer-2
Thank you so much for your time in reviewing the manuscript.
Comment 1: There is a 2004 version of RNI published by WHO/FAO. The RNI suggested for 12% bioavailability of iron are somewhat like the number shown in Table 2 of manuscript. However the RNI for adolescent girls is different. The link and related tables are provided below. Authors are suggested to clarify the suitable RNI used for analyzing data of this study.
Response 1: Thank you so much for pointing this. As per your (comment 5) and other reviewers comments, we have re-analyzed the nutrient intake data using EAR cut-points instead of RNI, in lines 119-123.
Comment 2: Since the age range for adolescent girls (13-18 yr) and adult women (19-49 yr) are different from the age groups specified by RNI of WHO, authors are suggested to explain in detail about how the NAR was calculated for adolescent girls. Since the status of deficiency for nutrients are different, adding a column of NAR for each nutrient in Table 2 between RNI and MAR is suggested.
Response 2: Thank you so much for asking this question. In cases where age groups of adolescent girls were different (e.g. 9-13 y and 14-18 y), we have averaged the requirements of age groups to determine the EAR values of our selected age group (13-18 y). Later, individual intake was divided by EAR in order to get NAR of a specific nutrient. You know NAR is the ratio of individual intake vs. EAR of a specific nutrient. Table 3 is formed with the prevalence of inadequate intake of a specific nutrient based on NAR values. A column with only NAR ratios may not make much sense. We have now added p values in EAR column (Table 2) to show the differences between mean intake vs. EAR, for the nutrients that had lower intakes.
Comment 3: Please explain how the nutrient intakes from 24 hr recalls were compared with RNI. Was it compared using individual data with RNI first, then calculated the average NAR for specific nutrient? Or was NAR derived by dividing average intake of each nutrient from 24 hr recalls by RNI.
Response 3: Thank you for asking this question. Individual intake data of each nutrient were compared with the age- and sex-specific EAR, in order to get NAR of each nutrient.
Comment 4: Since RNI = EAR x 2SD of EAR, and EAR (estimated average requirement) is around 70% of RNI(assume SD=12-15% SD), comparing average nutrient intake with RNI may overestimate the extent of nutrient deficiency. Authors may consider perform some more comparison of average nutrient intakes with EARs, or state this fact as a limitation of the study.
Response 4: Thank you so much for pointing this. We agree with your concern. In the revised version, we have used EAR cut-point instead of RNI and compared intake vs. EAR of the nutrients that are deficient.
Comment 5: The format of references is inconsistent and information of many references is incomplete. (References 1-3, 8, 12,24, 36, 54, 57, etc)Please check carefully.
Response 5: Thank you for the comment. We have revised the references.

Reviewer 3 Report
Overall, there are punctuation errors (too many commas) and some incomplete/confusing (lines 230-231) and run-on sentences (lines 52-56, 205-206, 239-243) . I found the “RA” abbreviation and use confusing and unnecessary since you have already described your population as being females of reproductive age.
Introduction
Pg 2, lines 56-61 and 68-71 are methods and instead should be placed in the methods section.
Materials and methods
Line 79, insert lines 257-260 re IRB approval. Was consent verbal or written?
2.1 dietary data collection – more information is needed for the reader to
Line 82 – include what the enumerators were trained on for the secondary analysis data collection; any inter-rater reliability?
Line 85 – I am unclear what a “preceding 24 hr recall method” is. Actually, as described I am having a hard time following exactly what the procedure was for collecting the 24 hr recall and determining portions consumed – please clarify
Line 86- “individual level” dietary intake – do you mean the participant? Replacing participant for individual would be clearer for who you are referring to. How did the participant estimate the food portion they recalled consuming?
Lines 88-89 – the equation is not clear to me
Raw weight of ingredients (household) is clear but not x amount – is this weight? Volume? or proportion?; same for denominator amount
Since I am not familiar with this methodology as stated, maybe an example will help? There is no reference for this method either.
2.3 calculating daily dietary nutrient intakes
Line 102 - use of RNI. I am most familiar to using the EAR cut point method, not RNI, for determining adequacy of a nutrient intake of a population. According to the authors reference, I quote from pg 5:
“There are several statistical approaches that can be used to estimate the prevalence of inadequate intakes from the distribution of intakes and requirements. One such approach the EAR cut-point method which defines the fraction of a population that consumes less than the EAR for a given nutrient. It assumes that the variability of individual intakes is at least as large as the variability in requirements and that the distribution of intakes and requirements are independent of each other. The latter is most likely to be true in the case of vitamins and minerals, but clearly not for energy. The EAR cut-point method requires a single population with a symmetrical distribution around the mean. If these conditions are met, the prevalence of inadequate intakes corresponds to the proportion of intakes that fall below the EAR. It is clearly inappropriate to examine mean values of population intake and RNI to define the population at risk of inadequacy. The relevant information is the proportion of intakes in a population group that is below the EAR, not below the RNI (4, 5).”
The authors, in my opinion, need to justify why they did not use the EAR cutpoint method for determining the proportion with inadequate intake or re-analyze.
Line 149 section dietary nutrient intakes…. Usually you do not report the exact same information in the text that is in the Table – please revise
Line 156-157 you state that dietary energy intake is higher than RNI, but do not provide statistical evidence of this. Please include p value showing significance
In this section, there is a missed opportunity to report on most commonly consumed foods contributing to adequate energy intake in relation to need and for foods providing specific nutrients which were more adequate in the diet, like niacin, B6, vit C, Zn or sources of limited nutrients.
Discussion
The format of the discussion is odd to me. Usually you first report your findings, next how those compare to other studies, then the implications or why that is important. Please revise.
Additionally you do not need to repeat p values of your findings if previously reported in results. For each of the nutrients you discuss, it would be helpful to include whether or not there are available food sources, what they are and could those be used meet these requirement and/or what limitations (season, cost, availability, etc.) might be.
Line 228 – “Poor dietary practices” sounds judgmental to me..might it be reduced access to affordable and nutrient dense foods? Please revise.
254-256 should be included in the discussion since use of only one 24 hr recall is a large limitation. You did not include any strengths of the study – please add.
Lines 236-243 The authors suggest a solution of using fish which is fine though narrow, and gives the appearance of a bias given the employment of two of the authors. This should be acknowledged in potential biases since CLB and SHT reviewed, edited and contributed to the manuscript.
Other factors should also be considered in your discussion and conclusions based on your data – education of WRA and men on diet diversity using what is available, keeping girls in school, delaying marriage and pregnancy.
Lines 242-243 – “especially if policies..” makes one wonder if the fisheries are cost-effective?
Conclusion
Suggest you be more precise and list the nutrients with high prevalence of inadequacy. And those that are higher prevalence in adolescents vs adults. Please also consider additional risk since adolescent requirements are higher.
It would be better if you could make a conclusion regarding the implications of your findings.
Author Response
Responses for reviewer-3
Thank you so much for the time in reviewing the manuscript
Comment 1: Overall, there are punctuation errors (too many commas) and some incomplete/confusing (lines 230-231) and run-on sentences (lines 52-56, 205-206, 239-243) . I found the “RA” abbreviation and use confusing and unnecessary since you have already described your population as being females of reproductive age.
Response 1: Thank you for the comments. We have revised lines 230-231 (now lines 256-258), 52-56 (now lines 55-59), 205-206 (now lines 237-238), 239-243 (now lines 267-272) as suggested. We have removed RA as suggested.
Comment 2: Pg 2, lines 56-61 and 68-71 are methods and instead should be placed in the methods section.
Response 2: Thank you for pointing this. Authors have moved the study objectives just before the method (lines 67-70) and reasonings were placed just before the objectives (lines 55-59).
Comment 3: Line 79, insert lines 257-260 re IRB approval. Was consent verbal or written?
Response 3: Thank you for pointing these. Line 79 has been removed. Verbal consent was added in line 81.
Comment 4: 2.1 dietary data collection – more information is needed for the reader to
Response 4: Thank you for pointing this. More information regarding dietary data collection is added in lines 92-102.
Comment 5: Line 82 – include what the enumerators were trained on for the secondary analysis data collection; any inter-rater reliability?
Response 5: More information regarding enumerators training and supervision including quality check has been added in lines 84-91.
Comment 6: Line 85 – I am unclear what a “preceding 24 hr recall method” is. Actually, as described I am having a hard time following exactly what the procedure was for collecting the 24 hr recall and determining portions consumed – please clarify
Response 6: Thank you for asking this question. This is 24 h dietary recall method that provides quantitative information on individual diets. We have revised the equation in line 101, hope it is more clearer now. We also added a reference (ref.28) now in line 94 of this 24 h recall method.
Comment 7: Line 86- “individual level” dietary intake – do you mean the participant? Replacing participant for individual would be clearer for who you are referring to. How did the participant estimate the food portion they recalled consuming?
Response 7: Thank you for asking this. Here individual means intra-household level (individuals in the household). We have revised the paragraph now in lines 92-98.
Comment 8: Lines 88-89 – the equation is not clear to me
Raw weight of ingredients (household) is clear but not x amount – is this weight? Volume? or proportion?; same for denominator amount
Since I am not familiar with this methodology as stated, maybe an example will help? There is no reference for this method either.
Response 8: Thank you for pointing this. Yes, you are right, amount means weight. This is like: raw weight of ingredients used to prepare foods or mixed dishes at the household*(times)cooked weight of foods or mixed dishes that an individual in the household consumed/(divide) cooked weight of foods or mixed dishes at the household. We have revised the equation in lines 101-102 and shared a reference in line 93.
Comment 9: 2.3 calculating daily dietary nutrient intakes
Line 102 - use of RNI. I am most familiar to using the EAR cut point method, not RNI, for determining adequacy of a nutrient intake of a population. According to the authors reference, I quote from pg 5:
“There are several statistical approaches that can be used to estimate the prevalence of inadequate intakes from the distribution of intakes and requirements. One such approach the EAR cut-point method which defines the fraction of a population that consumes less than the EAR for a given nutrient. It assumes that the variability of individual intakes is at least as large as the variability in requirements and that the distribution of intakes and requirements are independent of each other. The latter is most likely to be true in the case of vitamins and minerals, but clearly not for energy. The EAR cut-point method requires a single population with a symmetrical distribution around the mean. If these conditions are met, the prevalence of inadequate intakes corresponds to the proportion of intakes that fall below the EAR. It is clearly inappropriate to examine mean values of population intake and RNI to define the population at risk of inadequacy. The relevant information is the proportion of intakes in a population group that is below the EAR, not below the RNI (4, 5).”
The authors, in my opinion, need to justify why they did not use the EAR cutpoint method for determining the proportion with inadequate intake or re-analyze.
Response 9: Thank you so much for pointing this. We have re-analyzed the data using EAR cut-point method to assess micronutrient intakes, now in line 119-123.
Comment 10: Line 149 section dietary nutrient intakes…. Usually you do not report the exact same information in the text that is in the Table – please revise
Response 10: Thank you for pointing this. We have revised it as advised in line 169.
Comments 11: Line 156-157 you state that dietary energy intake is higher than RNI, but do not provide statistical evidence of this. Please include p value showing significance
Response 11: Thank you for the comment. We have renamed it as “requirement” instead of “RNI” in lines 175-177. We have now added p values as suggested in column EAR, in Table 2.
Comment 12: In this section, there is a missed opportunity to report on most commonly consumed foods contributing to adequate energy intake in relation to need and for foods providing specific nutrients which were more adequate in the diet, like niacin, B6, vit C, Zn or sources of limited nutrients.
Response 12: Thank you for the comment. This is an important point, but at this point we are not looking at the sources of these nutrients. Various studies have shown that 70% of energy is coming from the staple in the diet of Bangladeshi people.
Discussion
Comment 13: The format of the discussion is odd to me. Usually you first report your findings, next how those compare to other studies, then the implications or why that is important. Please revise.
Response 13: Thank you so much for the suggestion. We have revised the discussion part thoroughly as advised.
Comment 14: Additionally you do not need to repeat p values of your findings if previously reported in results. For each of the nutrients you discuss, it would be helpful to include whether or not there are available food sources, what they are and could those be used meet these requirement and/or what limitations (season, cost, availability, etc.) might be.
Response 14: Thank you for the suggestion. We have removed p values from the discussion as advised. In lines 242-244, we have referred studies that showed energy intake is related to heavy reliance on rice of Bangladeshi people. At this point we do not have the information (seasonality, cost, and availability) mentioned above.
Comment 15: Line 228 – “Poor dietary practices” sounds judgmental to me..might it be reduced access to affordable and nutrient dense foods? Please revise.
Response 15: Thank you for the comment. Sorry, this is a literature, we have revised it to make the statement more clearer in line 255-258.
Comment 16: 254-256 should be included in the discussion since use of only one 24 hr recall is a large limitation. You did not include any strengths of the study – please add.
Response 16: Thank you for the suggestion. You know 24 h recall is used to assess dietary patterns, food groups, or nutrient intakes. Mean intakes of foods and nutrients can be measured using a single 24HR, while assessing the "usual intake" of a population requires that repeat 24HRs are collected from a sub-sample of the study population. We have added the strength along with limitations in lines 288-294.
Comment 17: Lines 236-243 The authors suggest a solution of using fish which is fine though narrow, and gives the appearance of a bias given the employment of two of the authors. This should be acknowledged in potential biases since CLB and SHT reviewed, edited and contributed to the manuscript.
Response 17: Thank you for the comments. You know these statements are based on published data. We have revised lines 270 -272.
Comment 18: Lines 242-243 – “especially if policies..” makes one wonder if the fisheries are cost-effective?
Response 18: Thank you for pointing this. We have revised the statement and removed the policy part (lines 267-272).
Comment 19: Other factors should also be considered in your discussion and conclusions based on your data – education of WRA and men on diet diversity using what is available, keeping girls in school, delaying marriage and pregnancy.
Response 19: Thank you for the suggestion. We have discussed little bit about girls education related to diet diversity and diet quality in line 247-250. Keeping girls in the school is very important and could be a policy discussion which is beyond the scope of the study.
Conclusion
Comment 20: Suggest you be more precise and list the nutrients with high prevalence of inadequacy. And those that are higher prevalence in adolescents vs adults. Please also consider additional risk since adolescent requirements are higher.
Response 20: Thank for the suggestions. We have revised the conclusions as suggested in lines in 274-279.
Comment 21: It would be better if you could make a conclusion regarding the implications of your findings.
Response 21: Thank you so much for the suggestion. We have added a statement of policy implications of the study in lines 281-282.
